# Exploring Urban Green Spaces’ Effect against Traffic Exposure on Childhood Leukaemia Incidence

**DOI:** 10.3390/ijerph20032506

**Published:** 2023-01-31

**Authors:** Carlos Ojeda Sánchez, Javier García-Pérez, Diana Gómez-Barroso, Alejandro Domínguez-Castillo, Elena Pardo Romaguera, Adela Cañete, Juan Antonio Ortega-García, Rebeca Ramis

**Affiliations:** 1Guadalajara University Hospital, 19002 Guadalajara, Spain; 2Cancer and Environmental Epidemiology Unit, Department of Epidemiology of Chronic Diseases, National Center for Epidemiology, Instituto de Salud Carlos III (Carlos III Institute of Health), 28029 Madrid, Spain; 3Centre for Biomedical Research in Epidemiology & Public Health (CIBER Epidemiología y Salud Pública—CIBERESP), 28029 Madrid, Spain; 4National Center for Epidemiology, Instituto de Salud Carlos III (Carlos III Institute of Health), 28029 Madrid, Spain; 5Spanish Registry of Childhood Tumours (RETI-SEHOP), University of Valencia, 46010 Valencia, Spain; 6Pediatric Environmental Health Speciality Unit, Department of Paediatrics, Environment and Human Health (EH2) Lab., Institute of Biomedical Research, IMIB-Arrixaca, Clinical University Hospital Virgen de la Arrixaca, 30120 Murcia, Spain; 7European and Latin American Environment, Survival and Childhood Cancer Network (ENSUCHICA), 30120 Murcia, Spain

**Keywords:** urban green spaces, traffic exposure, environmental factors, childhood cancer, childhood leukaemia, incidence, spatial epidemiology

## Abstract

Background: Several environmental factors seem to be involved in childhood leukaemia incidence. Traffic exposure could increase the risk while urban green spaces (UGS) exposure could reduce it. However, there is no evidence how these two factors interact on this infant pathology. Objectives: to evaluate how residential proximity to UGS could be an environmental protective factor against traffic exposure on childhood leukaemia incidence. Methods: A population-based case control study was conducted across thirty Spanish regions during the period 2000–2018. It included 2526 incident cases and 15,156, individually matched by sex, year-of-birth, and place-of-residence. Using the geographical coordinates of the participants’ home residences, a 500 m proxy for exposure to UGS was built. Annual average daily traffic (AADT) was estimated for all types of roads 100 m near the children’s residence. Odds ratios (ORs) and 95% confidence intervals (95% CIs), UGS, traffic exposure, and their possible interactions were calculated for overall childhood leukaemia, and the acute lymphoblastic (ALL) and acute myeloblastic leukaemia (AML) subtypes, with adjustment for socio-demographic covariates. Results: We found an increment of childhood leukaemia incidence related to traffic exposure, for every 100 AADT increase the incidence raised 1.1% (95% CI: 0.58–1.61%). UGS exposure showed an incidence reduction for the highest exposure level, Q5 (OR = 0.63; 95% CI = 0.54–0.72). Regression models with both traffic exposure and UGS exposure variables showed similar results but the interaction was not significant. Conclusions: Despite their opposite effects on childhood leukaemia incidence individually, our results do not suggest a possible interaction between both exposures. This is the first study about the interaction of these two environmental factors; consequently, it is necessary to continue taking into account more individualized data and other possible environmental risk factors involved.

## 1. Introduction

Childhood leukaemia has become an important epidemiology issue for decades. It is the most common childhood cancer, representing more than a third of the total cases during this life period [1,2]. However, we only know nearly 10 per cent of its etiology [3]: current evidence suggests a combination of genetic susceptibility and environmental factors as its cause [4]. Some environmental factors as proximity to industrial installations and exposure to radon or crop fields have been related to it [5,6,7]. Among the possible environmental factors linked to childhood leukaemia, air pollution stands out in the urban context.

Nowadays, air pollution is one of the main environmental challenges for global public health [8]. Inside urban centres, traffic emissions are one of the most relevant sources. Motor vehicles emit many different substances, some of which as benzene, particulate matter with a diameter of less than 2.5 µm (PM_2.5_), or polycyclic aromatic hydrocarbons, are listed as carcinogenic by the International Agency for Research on Cancer (IARC) [9]. Such is that, benzene was related to a leukaemia risk increment occupationally among adults and with acute lymphoblastic leukaemia (ALL) in their offspring [10,11].

Traffic-related air pollution is one of the most studied environmental agents also related to children’s health. Some evidenced effects, as respiratory or allergic disorders in infant populations, have been related to this environmental factor [12]. Recent literature supports a link between traffic exposure and childhood leukaemia risk. Wei et al. indicated that exposure to some traffic components like benzene or NO_2_ during the second and third trimester of pregnancy could increase the risk of leukaemia [13]. Furthermore, among childhood leukaemia subtypes, acute myleoblastic leukaemia (AML) seems to be more associated with this exposure [14]. Before these last two studies were published, a systematic review performed by Filippini et al. found 29 eligible studies and pointed out the support for the association between exposure to benzene and childhood leukaemia risk [15].

As a consequence to this growing concern about the harmful effects of air pollution, many important cities have focused their urban plans on increasing urban green spaces (UGS). Simultaneously, under the hypothesis that these UGS could have environmental and health effects, research has been developed, even though there is not an established definition of UGS yet [16]. UGS’ environmental effects go from a noise pollution or a possible urban heat island effect reduction, thanks to the possibility of their physical qualities [17,18], to environmental pollutants reductions by different plant mechanisms as dispersion, adsorption, or decomposition of them by plants [19,20,21]. Studies published during this time have revealed some positive associations of these spaces [22]. In particular, some beneficial associations with them have been observed in infant mortality, childhood obesity, mental health disorders, or birth weight [23]. In the same line, we have observed a childhood leukaemia incidence reduction too [24].

With this in mind, where trees could reduce environmental pollutants, our objective was to evaluate if the UGS around the children’s residences could reduce the traffic pollution effect on childhood leukaemia incidence, taking into account the most common histologic subtype also (ALL and AML). For this purpose, we organized this paper in three different sections: firstly, we estimated the effect of traffic exposure for all the childhood population at risk; secondly, we estimated the effect of UGS exposure for children living in urban areas; and thirdly, we estimated the effect of both exposures combined and the possible interaction between them on childhood leukaemia incidence.

## 2. Materials and Methods Record

### 2.1. Study Design

We conducted a population-based case-control study of childhood leukaemia in Spain, covering the period 2000–2018. Incident cases were children, newborn to 14 years old, with leukaemia diagnoses born during the studied period. They were extracted from the Spanish Registry of Childhood Tumours (RETI-SEHOP), which collects information from hospitals’ paediatric oncology units all over Spain. Each record included the child’s home address when diagnosed, which gave us the opportunity to geocode every house to estimate the exposure to different environmental factors.

The control population was all Spanish newborns during the same period. These data came from the Spanish Statistical Office Birth Registry (Instituto Nacional de Estadistica, INE) [25]. We extracted the mother’s home address coordinates from the Birth Registry. To preserve participants’ anonymity, coordinates were given a 30-metre random error. Both cases and controls’ residences coordinates were projected into the ETRS89/UTM zone 30N (EPSG:25830) using QGIS^®^ version 3.4.4 software (Free Software Foundation, Inc., Boston, MA, USA).

The studied region did not include the whole country. The selection criteria for the regions was based on the reported coverage of the last report of the RETI-SEHOP and INE’s infant population statistics. According to both databases, these regions represent 72% of the national infant population with higher coverage cases recorded, over 85% of the paediatric cases [26] (Figure 1).

### 2.2. Traffic Exposure Measure

We estimated the traffic exposure measurement for each studied child using the annual average daily traffic (AADT). This measure represents the total volume of vehicles crossing a road over the course of a year divided by 365 days. AADT for Spanish road and street was provided by the Spanish Ministry of Public Works and we measured it for all roads 100 m around the children’s residence [27].

### 2.3. UGS Selection and Measurement

For UGS selection, we used the Spanish Land Use Information System (SIOSE) databases from 2005, 2011, and 2014 [28]. We selected spaces whose level specifications were linked directly to UGS. We followed the same method explained in our previous study to measure this exposition [24]. Considering our previous results, we selected a 500 m buffer distance around each subject. Furthermore, we categorized this exposure into 5 levels (quintiles) using the controls UGS values obtained, in which quintile 1 (Q1) represented the lowest level (reference group).

### 2.4. Sociodemographic Covariates

For the analysis, we included as potential confounders the following variables: sex, year of birth, socio-economic status (SES), and activity rate (AR). SES and AR were extracted from a census-tract level of the 2001 INE Census. SES combines information regarding the activity, professional situation, and occupation of the heads of families in each census tract, ranging from 0.46 (worst) to 1.57 (best) [25]. AR was defined as the ratio between the number of adult people working and the population equal or older than 16 years old or over in the census tract.

### 2.5. Statistical Analysis

Firstly, to estimate the relationship between traffic exposure and childhood leukaemia incidence, we fitted a mixed multiple unconditional logistic regression model and non-linear regression model, via generalized additive models (GAM). For this analysis, we used all possible cases and all newborn records in the studied regions.

Secondly, mixed multiple unconditional logistic regression models were fitted to estimate odds ratios (ORs) and 95% confidence intervals (95%CIs) associated with UGS exposure and our incidence target using those cases with residence in the urban areas, defined as municipalities with populations equal of superior to 20,000 inhabitants. As controls, we randomly selected newborns with maternal residence in the same cities; these were individually matched to cases in a 6:1 ratio by sex and year of birth.

Finally, we fitted mixed multiple unconditional logistic regression models in which we introduced both environmental factors and their interactions to estimate ORs and 95 CIs. We used the same population than in the previous analysis.

The fitted models for these objectives were adjusted by all the above-mentioned covariates as potential confounders. In addition, independent models for ALL, AML subtypes, and the sensitivity group were developed for the second and third objective of this study.

The sensitivity analysis had the purpose of assessing robustness to the results and only cases with identical birth and diagnosis address were included.

The statistical programs Microsoft Excel 2021^®^ (Microsoft Corporation, One Microsoft Way, Redmond, WA 98052-6399), R^®^ version 4.1.1, STATA^®^ version 16 (StataCorp LLC, 4905 Lakeway Drive, College Station, TX 77845 USA), and the geographic information system QGIS^®^ version 3.4.4 were used. The tests conducted in this paper were found to be statistically significant if the *p*-value was less than 0.05.

## 3. Results

### 3.1. Descriptive Analysis

Table 1 shows the children’s characteristics selected for goals 2 and 3. A total of 2526 cases and 15,156 controls were included. The most relevant group were ALL cases with 2015 subjects (79.8%), followed by AML with 401 cases (15.9%). Regarding the sex distribution, all subgroups presented a similar number of boys and girls. Nevertheless, as far as the age at diagnosis, AML cases were younger than ALL subtype cases and the overall sample.

Looking at the sensitivity analysis, we were able to identify at least two thirds of the cases (1737; 68.8%) with the same address at birth and diagnosis. The majority of their characteristics were similar to the general group of cases except their traffic exposure, which was higher.

### 3.2. First Objective: Traffic Exposition and Childhood Leukaemia Incidence

The logistic regression model showed an increase of the childhood leukemia incidence: for every 100 AADT, the incidence rises by 1.1% (95% CI: 0.58–1.61%). Results from the non-linear model, as seen in Figure 2, in which we counted in all possible cases and all newborn records, showed a growing trend starting from protective values to risk values at around 400 AADT. At 550 AADT value the effect of traffic exposure as a non-lineal variable started to be significantly linked with a childhood leukaemia incidence increment.

### 3.3. Second Objective: UGS Exposure and Childhood Leukaemia Incidence

Table 2 shows the ORs for total leukaemia, ALL, AML, and sensitivity group UGS exposure at 500 m. Model estimations showed a decreased tendency: higher UGS levels presented lower childhood leukaemia incidences with respect to the UGS exposure reference group. Looking at individual results, the lowest and statistically significant value for all samples was that associated with the highest level of UGS exposure, Q5 (OR = 0.63; 95% CI = 0.54–0.72). The ALL subtype showed similar results compare to overall cases.

Looking at the AML subtype and those children with the same address at birth and diagnosis moment, we saw some difference. Despite the AML subtype quintiles presenting lower OR values compared to Q1, they did not shape a trend. Its lowest value was that associated to Q3 (OR = 0.55; 95% CI = 0.40–0.77). Children whose addresses were the same at birth and diagnosis time presented similar results and trend compared to all samples but with slightly higher results.

### 3.4. Third Objective: Traffic and UGS Exposure Effects on Childhood Leukaemia Incidence

To explore this point, we selected those children living in urban areas whom AADT was equal or higher than 550. Secondly, traffic exposure was categorized into five levels using the controls traffic exposure values in which Q1 represented the lowest level and reference group.

Table 3 shows the ORs for this analysis. As we can see, traffic exposure was related with higher childhood leukaemia incidence rates for any of the developed models. They showed a growing tendency effect. Looking at the main childhood leukaemia group, the highest and statistically significant one was that associated with Q5, the highest level of traffic exposure (OR = 1.40; 95% CI = 1.12–1.76). Comparing these results with the ALL subtype and those children with no change address ones, we observed a small incidence increment. On the contrary, AML subtype results presented lower incidence values.

Despite no significant result of UGS exposure, its effect did not disappear. There seems to be slight similarities with the results obtained in Section 3.3: higher UGS exposure levels have lower incidence values compared to the lowest exposition group. Finally, no significant interaction between traffic exposure and UGS exposure was found in any of the developed models.

## 4. Discussion

In this paper, we investigated if UGS near children’s residences could reduce the traffic exposure effect on childhood leukaemia incidence. This research evidenced that, on one hand, traffic exposure increased the incidence of this infant pathology while, on the other hand, UGS was related to a reduction. Moreover, the effect for both exposure factors seemed to be stronger at higher exposure levels. Nevertheless, we could not stablish an interaction between them.

These results must be interpreted with caution because, to our knowledge, this is the first study that takes in consideration these two environmental agents together in childhood leukaemia incidence. Nevertheless, the individual results for each exposure factor are in agreement with the prior research.

In relation to traffic exposure, in this study, we started with the methodology developed by Tamayo et al. to estimate childhood leukaemia incidence [29]. For this occasion, we estimated the traffic exposure as a numeric and categorical variable in order to include different forms of the potential effect. As we have seen, its relationship with childhood leukaemia could be interpreted as a dose-response effect, since the highest ORs found were those at the highest levels of traffic exposure. A recent systematic review by Filippini et al. indicated a link between traffic-related air pollution, particularly exposure to benzene, which is known to be a highly carcinogenic agent, and an increased risk of childhood leukaemia [15,30].

Traffic exposure has been an important environmental issue studied within the last decades. It is associated, among other things, with allergic diseases, lung disorders such as asthma or lung function reduction [12,31,32]. The methodology applied to measure this pollutant is quite heterogeneous. Different studies related it to measuring the NO, NO_2_, benzene, or PM_2.5_ exposition level or estimating traffic density close to subjects’ residence [29,33]. Time window exposition was also contrasted [34]. Although many others potential environmental factors, such as industrial pollution or indoor house pollutants, were included as confounders in this association, the possible role of UGS has not been considered until now.

Regarding our second exposure factor, increasing UGS has been stablished as one fundamental goal to improve human health, with actions such as heat and pollution effects reduction produced by traffic and buildings in city centres [35]. Greenspace impact on health has been supported consistently. Specifically, UGS has been related to beneficial children outcomes [36,37,38,39]. Pathways linking to these gains could be organized in three domains which are suggested to act together [40]. By extrapolating our methodology used previously [24] for the rest of the provinces included, the results obtained in this study reinforced the idea that UGS could be connected to a childhood leukaemia incidence reduction. The available evidence is gradually expanding, so it is necessary to identify more reliable mediators between UGS and leukaemia risk. Nevertheless, we could not demonstrate that these urban spaces mitigate the harmful effect of traffic exposure over leukaemia. This lack of interaction could also point to different mediating mechanisms. Exposure to UGS seems to modulate the anticancer immune response by increasing mental well-being by reducing cortisol levels, increasing production of natural killer cells by exposure to phytoncides, and promoting healthy lifestyles, meanwhile decreasing exposure to anthropogenic carcinogens [41,42].

Down below we list some limitations that should be taken into account when interpreting these results. Firstly, despite plants being able to reduce environmental pollutants thanks to its physiological characteristics [43,44], as we have worked with land cover maps, we could not take into account which plants types compound each UGS neither their health condition. Secondly, we did not introduce other possible environmental confounders like blue spaces. There is some evidence that blue spaces could reduce childhood leukaemia incidence also [45]. Although its mechanism is not stablished yet, it hypothesized that they could act in a similar way as green spaces [46]. In this occasion, we did not consider to introduce them in the regression models due to the cities’ characteristics involved in the study. The presence of these spaces is not similar in inland cities than coastal ones. Thirdly, we have worked with two national childhood registries (RETI-SEHOP and Spanish Statistical Office Birth Registry) which have a huge amount of subjects recorded but they do not include possible indoor or parental exposition and lifestyles data that could act as potential confounders [3,7,47]. In addition, it was not possible to incorporate any information or data regarding the time spend or the activities they engaged in while interacting with UGS

However, we tried to overcome some of these problems with a consistent number of cases recorded along the country from 2000 to 2018. This has allowed us to confirm previous traffic exposure and UGS results. Moreover, we have included two sociodemographic covariates, SES and activity rate. SES is a factor related to childhood cancer or greenspaces that has been studied and used as a potential modifier [48,49]. In a previous study, we could observe that it was related with low childhood leukaemia incidence [24].

Finally, we tried to reinforce our study by including a sensitivity analysis to explore the characteristics of those children whose home addresses were the same at birth and at diagnosis time. According to official Spanish data, the likelihood of a child changing their province of residence is estimated to be only 1% [50]. We could identify 1737 cases with the same address at birth and at diagnosis, 68.8% of cases. The results obtained from their models were close to the general case group so we could not identify any possible difference between both groups.

## 5. Conclusions

The purpose of the present research was to investigate if UGS exposure could reduce the traffic exposure effect on childhood leukaemia incidence. We have confirmed previous findings about traffic and UGS exposure effects on this infant disease individually, nonetheless we found no evidence of their interaction. Despite this lack of interaction, the reduction of traffic exposure and the increase of UGS exposure seem to be a worthy approach in order to reduce childhood leukaemia.

## Figures and Tables

**Figure 1 ijerph-20-02506-f001:**
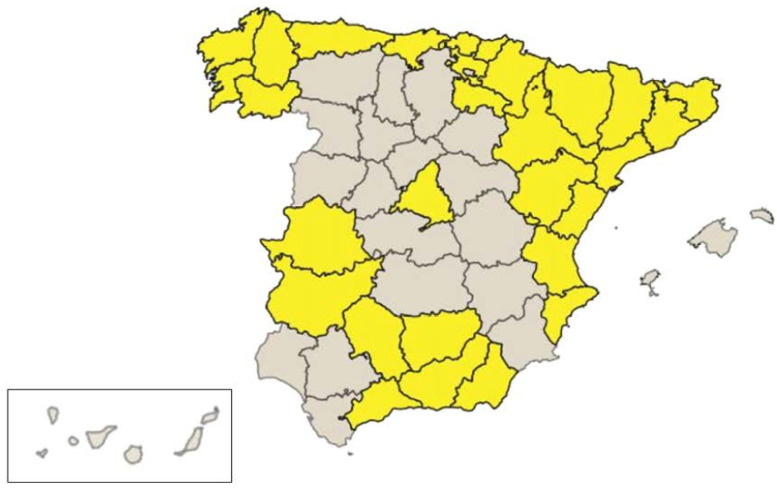
Spanish provinces selected for the study.

**Figure 2 ijerph-20-02506-f002:**
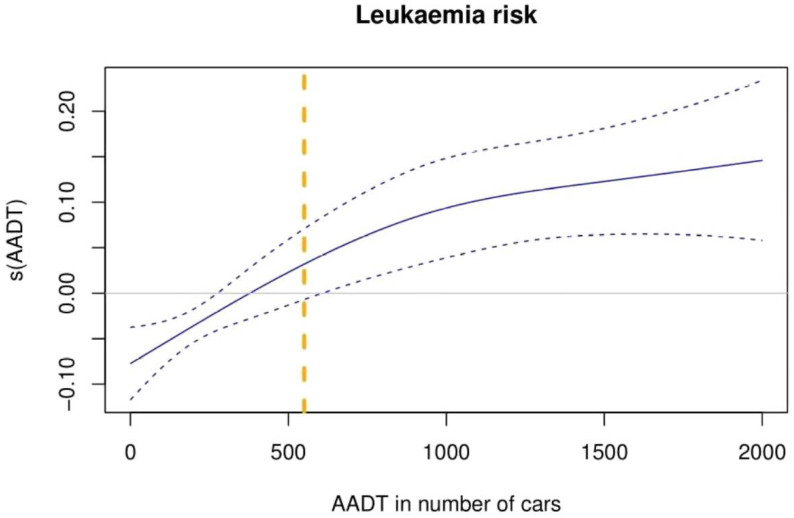
Overall childhood leukaemia exposure to traffic, measured in terms of AADT. Graphical representation of adjusted non-linear regression model for sex, year of birth, SES, and AR. s(AADT) represents AADT smooth effect on childhood leukaemia incidence. Blue dashed lines represent lower and upper 95% CI. Orange dashed line represents 550 AADT on the *x*-axis.

**Table 1 ijerph-20-02506-t001:** Childhood leukaemia—cases and sensitivity group and controls’ characteristics.

Characteristics	Controls (*n* = 15,156)	Cases (*n* = 2526)	ALL (*n* = 2015)	*p* Value ^a^	AML(*n* = 401)	*p* Value ^a^	Same Address (*n* = 1737)	*p* Value ^a^
Sex, *n* (%)								
Boy	9346 (55.1%)	1391 (55.1%)	1118 (55.5%)		217 (54.1%)		937 (56.0%)	
Girl	6810 (44.9%)	1135 (44.9%)	897 (44.5%)	0.802 ^#^	184 (45.9%)	0.763 ^#^	764 (44.0%)	0.561 ^#^
Age at diagnosis, median (IQR)	X	4 (5)	4 (4)	0.126 ^¥^	3 (6)	<0.001 ^¥^	4 (5)	0.574 ^¥^
Activity rate, mean (SD)	76.61 (5.29)	76.33 (5.50)	76.2 (5.6)	0.417 *	76.7 (5.0)	0.218 *	76.6 (5.34)	0.073 *
SES, mean (SD)	1.03 (0.14)	1.04 (0.13)	1.04 (0.14)	0.706 *	1.04 (0.13)	0.785 *	1.04 (0.13)	0.748 *
AADT, mean (SD)	698 (650)	640 (650)	632 (645)	0.685 *	661 (658)	0.555 *	714 (679)	<0.001 *
Histologic subtype, *n* (%)								
ALL	x	2015 (79.8%)					1375 (79.2%)	
AML	x	401 (15.9%)					287 (16.5%)	
CML	x	28 (1.1%)					19 (1.1%)	
Other specific leukaemia	x	48 (1.9%)					31 (1.8%)	
Non-specific leukaemia	x	34 (1.3%)					25 (1.4%)	0.978 ^#^

^a^ *p* value from main group of cases compared to leukaemia subtypes and sensitivity group. ^#^ Chi-square test for categorical variables, * Student’s *t*-test, ^¥^ Wilcoxon’s test. Abbreviations: SD—standard deviation, ALL—acute lymphoblastic leukaemia, AML—acute myeloblastic leukaemia, CML—chronic myeloblastic leukaemia, IQR—interquartile range.

**Table 2 ijerph-20-02506-t002:** Results of the UGS analysis for overall childhood leukaemias, ALL and AML subtypes, and identified cases with same address at birth and diagnosis. Adjusted models for sex, year of birth, SES, and activity rate.

	Childhood Leukaemias		ALL Subtype		AML Subtype		Same Address	
	Cases/Controls (*n*)	Adjusted OR (95% CI)	Cases/Controls (*n*)	Adjusted OR (95% CI)	Cases/Controls (*n*)	Adjusted OR (95% CI)	Cases/Controls (*n*)	Adjusted OR (95% CI)
UGS.Q1	658/3031	1	523/3031	1	105/3031	1	394/3031	1
UGS.Q2	497/3031	0.74 (0.65–0.84)	407/3031	0.77 (0.67–0.88)	76/3031	0.71 (0.52–0.96)	364/3031	0.91 (0.78–1.06)
UGS.Q3	464/3032	0.69 (0.61–0.78)	385/3032	0.73 (0.63–0.84)	60/3032	0.55 (0.40–0.77)	337/3032	0.84 (0.72–0.98)
UGS.Q4	482/3030	0.72 (0.63–0.82)	364/3030	0.69 (0.59–0.80)	93/3030	0.85 (0.64–1.14)	350/3030	0.87 (0.74–1.01)
UGS.Q5	425/3032	0.63 (0.54–0.72)	336/3032	0.63 (0.54–0.74)	67/3032	0.61 (0.44–0.84)	292/3032	0.72 (0.61–0.85)

OR and 95% CI for each quintile (Q) compare to the reference group (Q1 = lowest exposure to UGS). Abbreviations: UGS—urban green space, OR—odds ratio, CI—confidence interval, ALL—acute lymphoblastic leukaemia, AML—acute myeloblastic leukaemia, SES—socioeconomic status.

**Table 3 ijerph-20-02506-t003:** Traffic exposure and UGS exposure ORs extracted from the final adjusted regression model for overall childhood leukaemia, ALL and AML subtypes, and identified cases with same address at birth and diagnosis results. Adjusted models for sex, year of birth, SES, and activity rate.

	Childhood Leukaemias		ALL Subtype		AML Subtype		Same Address	
Quintile	Cases/Controls (*n*)	Adjusted OR (95% CI)	Cases/Controls (*n*)	Adjusted OR (95% CI)	Cases/Controls (*n*)	Adjusted OR (95% CI)	Cases/Controls (*n*)	Adjusted OR (95% CI)
TE.Q1	159/1359	1	120/1359	1	28/1359	1	110/1359	1
TE.Q2	212/1359	1.34 (1.08–1.67)	173/1359	1.45 (1.14–1.85)	33/1359	1.17 (0.71–1.97)	152/1359	1.38 (1.07–1.79)
TE.Q3	197/1360	1.25 (1.00–1.57)	164/1360	1.38 (1.08–1.78)	26/1360	0.93 (0.54–1.59)	160/1360	1.46 (1.13–1.89)
TE.Q4	195/1358	1.25 (1.00–1.57)	147/1358	1.26 (0.98–1.62)	36/1358	1.27 (0.77–2.12)	155/1358	1.41 (1.10–1.84)
TE.Q5	216/1360	1.40 (1.12–1.76)	169/1360	1.47 (1.14–1.89)	36/1360	1.23 (0.76–2.12)	177/1360	1.63 (1.26–2.10)
UGS.Q1	195/1359	1	156/1359	1	33/1359	1	149/1359	1
UGS.Q2	190/1359	0.95 (0.77–1.18)	151/1359	0.95 (0.75–1.20)	28/1359	0.84 (0.50–1.41)	151/1359	0.99 (0.78–1.26)
UGS.Q3	215/1359	1.07 (0.87–1.32)	170/1360	1.06 (0.84–1.34)	33/1359	0.99 (0.60–1.63)	163/1359	1.07 (0.84–1.36)
UGS.Q4	199/1359	0.99 (0.80–1.23)	155/1358	0.97 (0.77–1.24)	36/1359	1.07 (0.65–1.75)	161/1359	1.05 (0.83–1.34)
UGS.Q5	180/1360	0.90 (0.72–1.12)	141/1360	0.89 (0.70–1.14)	29/1360	0.87 (0.51–1.46)	130/1360	0.86 (0.66–1.10)

OR and 95% CI for each quintile (Q) compare to the reference group (Q1 = lowest exposure). Abbreviations: TE—traffic exposure, UGS –urban green space, OR—odds ratio, CI—confidence interval, ALL—acute lymphoblastic leukaemia, AML—acute myeloblastic leukaemia, SES—socioeconomic status.

## Data Availability

The data presented in this study are available on request from the corresponding author. The data are not publicly available due to keep the subject’s privacy.

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
