# Peer review of "Exploring Urban Green Spaces’ Effect against Traffic Exposure on Childhood Leukaemia Incidence"

_ijerph, 2023, doi:10.3390/ijerph20032506_

Round 1
Reviewer 1 Report
1. Based on the results of this work, no correlation between urban green space and the incidence of childhood leukemia was clearly derived. Generally speaking, urban green spaces, with natural and artificial vegetation as the main existence form of urban land, has health-promoting effects such as alleviating environmental stresses such as air pollution, noise and urban heat island effect; providing comfortable space, alleviating people’s mental stress and relieving mental fatigue. Especially in the context of the COVID-19 epidemic, UGS plays an important function in the public health quality of urban space. In this respect, is it possible that UGS may reduce the incidence of childhood leukemia rather than indirectly increase the risk of childhood leukemia incidence through traffic exposure? On what considerations do the authors base their analysis of this paradox?2. Please improve the quality of the graphs in this paper.
Author Response
Comment 1: Based on the results of this work, no correlation between urban green space and the incidence of childhood leukemia was clearly derived. Generally speaking, urban green spaces, with natural and artificial vegetation as the main existence form of urban land, has health-promoting effects such as alleviating environmental stresses such as air pollution, noise and urban heat island effect; providing comfortable space, alleviating people’s mental stress and relieving mental fatigue. Especially in the context of the COVID-19 epidemic, UGS plays an important function in the public health quality of urban space. In this respect, is it possible that UGS may reduce the incidence of childhood leukemia rather than indirectly increase the risk of childhood leukemia incidence through traffic exposure? On what considerations do the authors base their analysis of this paradox?
Response 1: In this study we have tried to analyses how UGS and traffic exposure could be related with childhood leukaemia incidence. For this occasion, we tried to explore individually each agent firstly (first and second objectives). As we have seen traffic exposure increased childhood leukaemia incidence meanwhile UGS exposure reduced it.
According to the results obtained from the third objective in which we introduced both factors in the same model with their interaction, we have observed that the effects of them followed the same results as the previous but we cannot stablish that they are linked because there was no significant interaction.
Comment 2: Please improve the quality of the graphs in this paper.
Response 2: we have improved the quality of the graphs. In addition, they are submitted again in case of a resolution problem.

Reviewer 2 Report
Need few improvement based on the following comments:
Line 20 - environmental 'agent' is misleading. Use 'barrier' or 'protection'
Line 25 - ratio of the case: control is more than 4. The result may be in favor of the control group which leads to misinterpretation. Obviously, the large crowd will be in favor.
Line 33 - Odd ratio is NOT significant as 95%CI is NOT significant. Misleading statement by the author.
Line 38 - CONCLUSION is not correct due to Line 33 above.
Line 48 - start with Reference 3? Where are Ref 1 & 2?
Line 95 - children aged '0' to..... Instead of '0', use newborn
Line 108 - method of selection is not clear
Line 117 - which year?
Line 128 - selection of Q1 as reference is not well explained
Line 136 - please state the significant cut-off point for p-value.
Line 171 - Table 1: missing data for 'Same address' (n=1737) and % should not be 100%. Need to explain the missing data in the text. The objective should include an analysis of subtypes of leukemia too.
Line 176 - 'insignificant increase'.. The first objective is not supported by statistics.
Line 201 - Table 2: change 'REFERENCE' to 1. The objective should include an analysis of subtypes of leukemia.
Line 223 - Table 3: use 1 instead of REFERENCE. The objective should include an analysis of subtypes of leukemia.
Line 307 - 'reduction of traffic exposure' is an insignificant finding.
Author Response
Comment 1: Line 20 - environmental 'agent' is misleading. Use 'barrier' or 'protection'
Response 1: we have changed it.
Comment 2: Line 25 - ratio of the case: control is more than 4. The result may be in favor of the control group which leads to misinterpretation. Obviously, the large crowd will be in favor.
Response 2: For our first objective we decided to include all possible subjects because we wanted to obtain a realistic traffic exposure effect. For the following objectives, those developed inside urban centers, we decided to choose the ratio 6:1 and not 4:1 because the exposure is approximated by the spatial distribution of the children and the green spaces. We needed a big control group that could gave us a better picture of the population density (children density) along the studied area.
Comment 3: Line 33 - Odd ratio is NOT significant as 95%CI is NOT significant. Misleading statement by the author.
Response 3: Thank you for point it us. It can be confused. We have corrected this line and lines 153-154 to clarify this relationship.
Comment 4: Line 38 - CONCLUSION is not correct due to Line 33 above.
Response 4: According to the previous response, this mistake is fixed.
Comment 5: Line 48 - start with Reference 3? Where are Ref 1 & 2?
Response 5: They were deleted. We have corrected it.
Comment 6: Line 95 - children aged '0' to..... Instead of '0', use newborn
Response 6: This phrase has been rewritten.
Comment 7: Line 108 - method of selection is not clear
Response 7: The urban green spaces were selected from the Spanish Land Use Information System (SIOSE) databases. This information is available in the media resources section of the Spanish National Geographic Institute (IGN). This selection method is explained carefully in our previous study.
- Ojeda Sanchez et al. 2021. Urban green spaces and childhood leukemia incidence: A population-based case-control study in Madrid. Environ Res 2021 Nov;202:111723. doi: 10.1016/j.envres.2021.111723. Epub 2021 Jul 19.
Comment 8: Line 117 - which year?
Response 8: Both sociodemographic covariates were extracted from the 2001 INE Census (See line 115).
Comment 9: Line 128 - selection of Q1 as reference is not well explained
Response 9: We have rewritten this paragraph in order to clarify how we have determined Q1 as the reference group. This paper follows the same method used in our previous one about childhood leukaemia incidence and urban green spaces.
Comment 10: Line 136 - please state the significant cut-off point for p-value.
Response 10: we have included p-value cut-off point at the end of Statistical analysis point (subsection 2.5). It was stablished in 0.05.
Comment 11: Line 171 - Table 1: missing data for 'Same address' (n=1737) and % should not be 100%. Need to explain the missing data in the text. The objective should include an analysis of subtypes of leukemia too.
Response 11: We have identified 68.8% of cases whose addresses were the same at birth time and diagnosis. The proportions showed in “same address” column in Table 1 was related with the group of cases whose address were not modified between birthtime and childhood leukaemia diagnoses. This paper included as an objective the analyses of the most incident leukaemia subtypes ALL and AML. This objective was stablished in line 75 and their results were shown along this paper (subsection 3.3. and 3.4.).
Comment 12: Line 176 - 'insignificant increase'.. The first objective is not supported by statistics.
Response 12: For the first objective, we fitted a nonlinear regression model. Figure 2 shows the result. The graph shows the change in the effect of traffic over leukemia incidence, with the change in number of cars: blue line. The blue dashed lines represent the lower and upper 95%CI (statistic). When the lines are above cero, this means the risk increased. Therefore, when the dashed lines are above cero, we can say that the effect is statistically significant. We haven’t use the traditional p-value for this analysis because it changes with the changes in the number of cars.
Comment 13: Line 201 - Table 2: change 'REFERENCE' to 1. The objective should include an analysis of subtypes of leukemia.
Response 13: We have changed “Reference” to 1. The objective includes an analysis of the most incidence subtypes of leukaemia as we have set in line 76 (ALL subtype and AML subtype).
Comment 14: Line 223 - Table 3: use 1 instead of REFERENCE. The objective should include an analysis of subtypes of leukemia.
Response 14: We have changed “Reference” to 1. The objective includes an analysis of the most incidence subtypes of leukaemia as we have set in line 76 (ALL subtype and AML subtype).
Comment 15: Line 307 - 'reduction of traffic exposure' is an insignificant finding.
Response 15: We are not agree with it. As we can see in subsection 3.2. lower AADT values could be related with less childhood leukaemia incidence. In addition, our results are in line with the previous articles published about traffic exposure and childhood leukaemia. The last conclusion section’s sentence tries to encourage that a reduction of the traffic exposure could reduce childhood leukaemia incidence.
